# The Unreasonable Effectiveness of Fully-Connected Layers for Low-Data Regimes

**Peter Kocsis**
Technical University of Munich
`peter.kocsis@tum.de`

**Peter Súkeník**
Technical University of Munich
`peter.sukenik@trojsten.sk`

**Guillem Brasó**
Technical University of Munich
`guillem.braso@tum.de`

**Matthias Nießner**
Technical University of Munich
`niessner@tum.de`

**Laura Leal-Taixé**
Technical University of Munich
`leal.taixe@tum.de`

**Ismail Elezi**
Technical University of Munich
`ismail.elezi@tum.de`

[peter-kocsis.github.io/LowDataGeneralization/](peter-kocsis.github.io/LowDataGeneralization/)

## Abstract

Convolutional neural networks were the standard for solving many computer vision tasks until recently, when Transformers of MLP-based architectures have started to show competitive performance. These architectures typically have a vast number of weights and need to be trained on massive datasets; hence, they are not suitable for their use in low-data regimes. In this work, we propose a simple yet effective framework to improve generalization from small amounts of data. We augment modern CNNs with fully-connected (FC) layers and show the massive impact this architectural change has in low-data regimes. We further present an online joint knowledge-distillation method to utilize the extra FC layers at train time but avoid them during test time. This allows us to improve the generalization of a CNN-based model without any increase in the number of weights at test time. We perform classification experiments for a large range of network backbones and several standard datasets on supervised learning and active learning. Our experiments significantly outperform the networks without fully-connected layers, reaching a relative improvement of up to $16\%$ validation accuracy in the supervised setting without adding any extra parameters during inference.

## 1 Introduction

Convolutional neural networks (CNNs) [1, 2] have been the dominant architecture in the field of computer vision. Traditionally, CNNs consisted of convolutional (often called cross-correlation) and pooling layers, followed by several fully-connected layers [3–5]. The need for fully-connected layers was challenged in an influential paper [6], and recent modern CNN architectures [7–11] discarded

36th Conference on Neural Information Processing Systems (NeurIPS 2022).

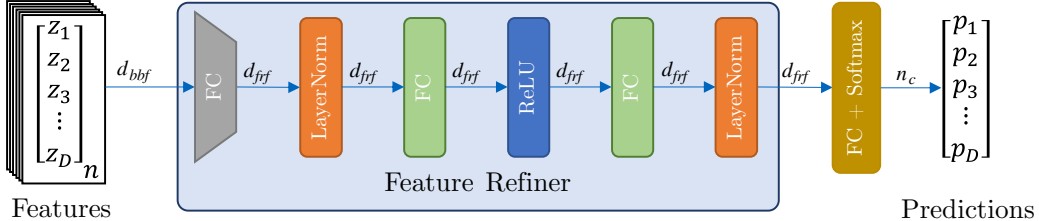

Figure 1: **Feature Refiner architecture.** Our network takes the features extracted by the backbone network. We apply dimension-reduction to reduce the model parameters followed by a symmetric 2-layered multi-layer perceptron.

them without a noticeable loss of performance and a drastic decrease in the number of trainable parameters.

Recently, the "reign" of all-convolutional networks has been challenged in several papers [12–16], where CNNs were either replaced (or augmented) by vision transformers or replaced by multi-layer perceptrons (MLPs). These methods remove the inductive biases of CNNs, leaving more learning freedom to the network. While showing competitive performance and often outperforming CNNs, these methods come with some major disadvantages. Because of their typically large number of weights and the removed inductive biases, they need to be trained on massive datasets to reach top performance. As a consequence, this leads to long training times and the need for massive computational resources. For example, MLPMixer [14] requires a thousand TPU days to be trained on the ImageNet dataset [17].

In this paper, instead of entirely replacing convolutional layers, we go back to the basics, and augment modern CNNs with fully-connected neural networks, combining the best of both worlds. Contrary to new alternative architectures [14, 15] that usually require huge training sets, we focus our study on the opposite scenario: the low-data regime, where the number of labeled samples is very-low to moderately low. Remarkably, adding fully-connected layers yields a significant improvement in several standard vision datasets. In addition, our experiments show that this is agnostic to the underlying network architecture and find that fully-connected layers are required to achieve the best results. Furthermore, we extend our study with two other settings that typically deal with a low-data regime: active and semi-supervised learning. We find that the same pattern holds in both cases.

An obvious explanation for the performance increase would be that adding fully-connected layers largely increases the number of learnable parameters, which explains the increase in performance. To disprove this theory, we use knowledge distillation based on a gradient gating mechanism that reduces the number of used weights during inference to be equal to the number of weights of the original networks, e.g., ResNet18. We show in our experiments that this reduced network achieves the same test accuracy as the larger (teacher) network and thus significantly outperforms equivalent architecture that does not use our method.

In summary, our **contributions** are the following:

- We show that adding fully-connected layers is beneficial for the generalization of convolutional networks in the tasks working in the low-data regime.
- We present a novel online joint knowledge distillation method (OJKD), which allows us to utilize additional final fully-connected layers during training but drop them during inference without a noticeable loss in performance. Doing so, we keep the same number of weights during test time.
- We show state-of-the-art results in supervised learning and active learning, outperforming all convolutional networks by up to $16\%$ in the low data regime.

## 2   Methodology

We propose a simple yet effective framework for improving the generalization from a small amount of data. In our work, we bring back fully-connected layers at the end of CNN-based architectures.

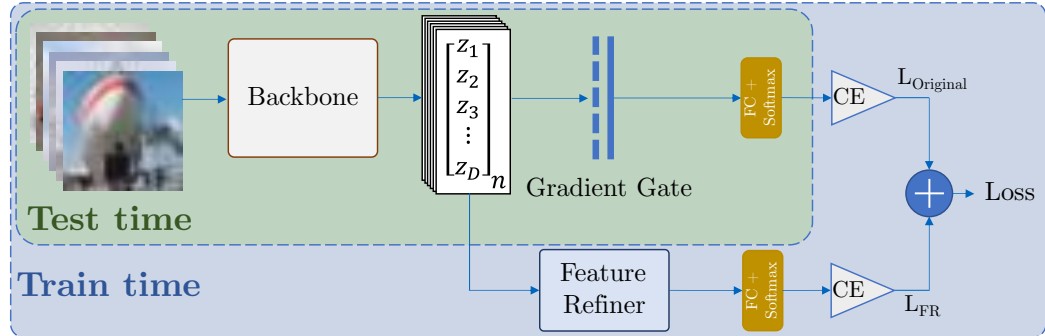

Figure 2: **Online Joint knowledge distillation pipeline.** Besides the baseline network's classification head, we append an extra head with our Feature Refiner. The network is trained with the composed of the two heads' cross-entropy loss. During training, the gradients coming from the single fully-connected layer are blocked. As a result, the backbone is updated only by the second head with our Feature Refiner. In test time, this extra head can be neglected without any noticeable performance loss.

We show that by adding as little as $0.37\%$ extra parameters during training, we can significantly improve the generalization in the low-data regime. Our network architecture consists of two main parts: a convolutional backbone network and our proposed Feature Refiner (FR) based on multi-layer perceptrons. Our method is task and model-agnostic and can be applied to many convolutional networks.

In our method, we extract features with the convolutional backbone network. Then, we apply our FR followed by a task-specific head. More precisely, we first reduce the feature dimension $d_{bbf}$ to $d_{frf}$ with a single linear layer to reduce the number of extra parameters. Then we apply a symmetric two-layer multi-layer perceptron wrapped around by normalization layers. We present the precise architecture of our Feature Refiner in Figure 1.

## 2.1 Online Joint Knowledge Distillation

One could argue that using more parameters can improve the performance just because of the increased expressivity of the network. To disprove this argument, we develop an online joint knowledge distillation (OJKD) method. Our OJKD enables us to use the exact same architecture as our baseline networks during inference and utilizes our FR solely during training.

We base our training pipeline on the baseline network's architecture. We split the baseline network into two parts, the convolutional backbone for feature extraction and the final fully-connected classification head. We append an additional head with our Feature Refiner. We devise the final loss as the sum of the two head's losses, making sure both heads are trained in parallel (online) and enforcing that they share the same network backbone (joint). During inference, we drop the additional head and use only the original one, resulting in the exact same test time architecture as our baseline. In other words, our FR head is the teacher network that shares the backbone with the student original head, and we distill the knowledge of the teacher head into the student head.

However, the key ingredient of our OJKD is the gradient-gating mechanism; we call it Gradient Gate (GG). This gating mechanism blocks the gradient of the original head during training, making the backbone only depend on our FR head. We implement this functionality with a single layer. During the forward pass, GG works as identity and just forwards the input without any modification. However, during the backward pass, it sets the gradient to zero. This way, the original head's gradients are backpropagated only until the GG. While the original head gets optimized, it does not influence the training of the backbone but only adapts to it. Consequently, the backbone is trained only with the gradients of our FR head. Furthermore, the original head can still fit to the backbone and reach a similar performance as the FR head. We find that we can still improve upon the baseline without our gating mechanism but reach lower accuracy than when we use it. We show the pipeline of our OJKD in Figure 2.

# 3 Experiments

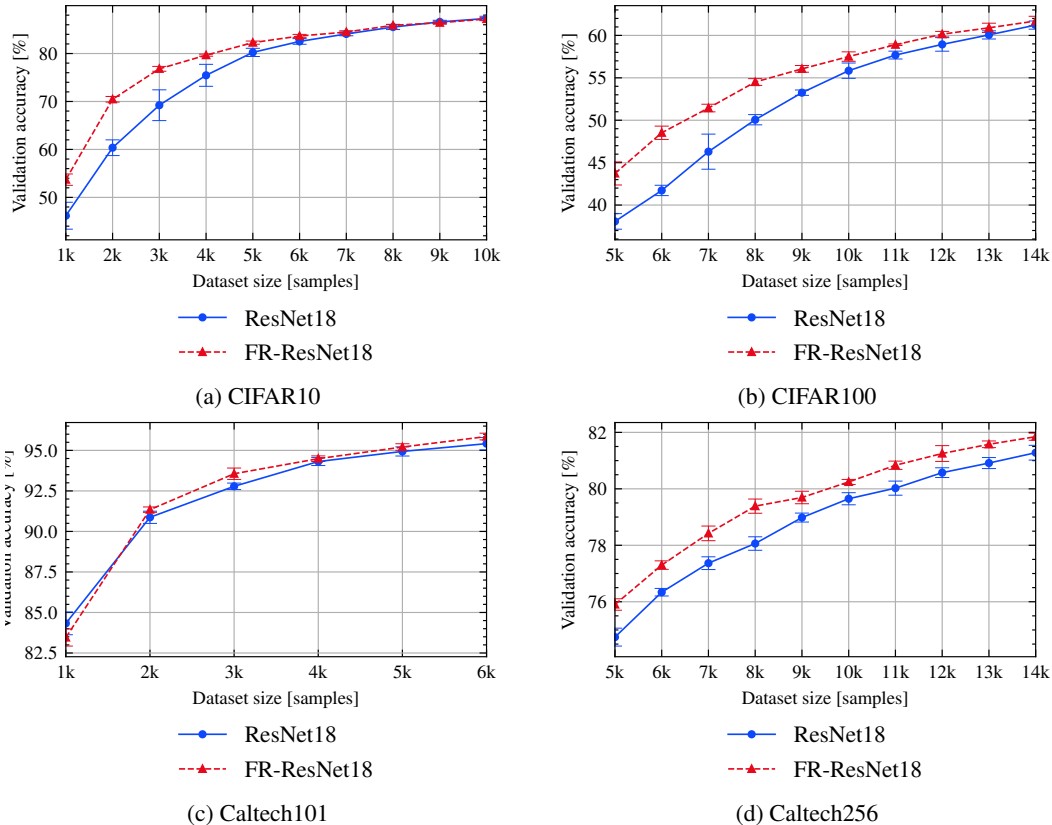

Figure 3: **Comparisons with ResNet18.** We compare our approach (FR) to the baseline network ResNet18 in supervised learning. Our method significantly outperforms the baseline, especially in the more challenging earlier stages, when we have a small amount of data.

In this section, we demonstrate the substantial effectiveness of our simple approach improving the performance of neural networks in low-data regimes.

**Datasets and the number of labels.** For all experiments, we report accuracy as the primary metric and use four public datasets: CIFAR10 [18], CIFAR100 [18], Caltech101 [19], and Caltech256 [19]. We use the predefined train/test split for the CIFAR datasets, while we split the Caltech datasets into 70% training and 30% testing, maintaining the class distribution. In the simpler datasets, CIFAR10 and Caltech101, we start with an initial labeled pool of 1000 images, while in the more complicated, CIFAR100 and Caltech256, we start with 5000 labeled images. In both cases, we incrementally add 1000 samples to the labeled pool in each cycle and evaluate the performance with the larger and larger training datasets. We use the active learning terminology for a 'cycle', where a cycle is defined as a complete training loop.

**CNN backbones.** For most experiments, we use ResNet18 [8] as our backbone ($d_{bbf} = 512$), and reduced feature size $d_{frf} = 64$. We compare the results of our method with those of pure ResNet18 on both the supervised and active learning setups. Note that the supervised case (Figure 3) is equivalent to a random labeling strategy in an active learning setup. We compare our method with various active learning strategies. We also compare to a non-convolutional network, the MLPMixer [14]. Finally, to show the generability of our method, we also experiment with other backbone networks: ResNet34, EfficientNet, and DenseNet. We run each experiment 5 times and report the mean and standard deviation. We train each network in a single GPU. We summarize the results in plots and refer to our supplementary material for exact numbers and complete implementation details.

**Training details.** For the CIFAR experiments, we follow the training procedure of [20]. More precisely, we train our networks for 200 epochs using SGD optimizer with learning rate 0.1, momentum

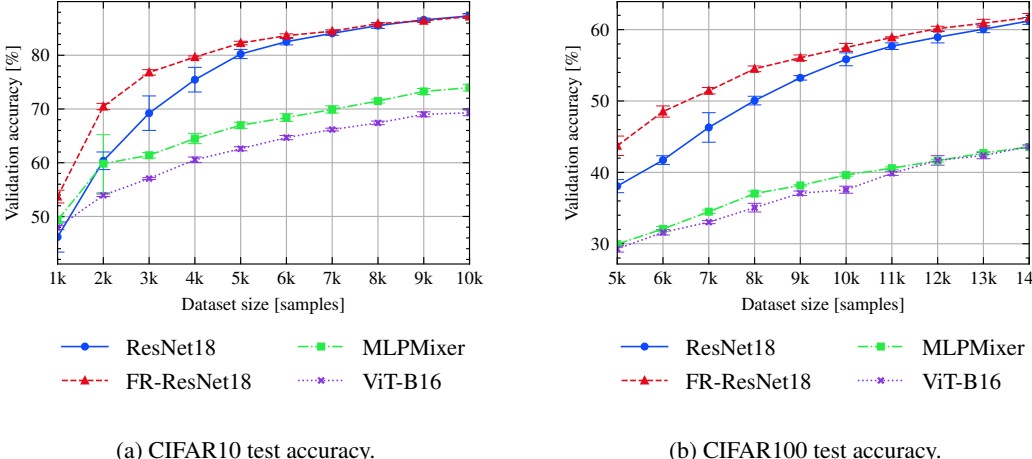

(a) CIFAR10 test accuracy.

(b) CIFAR100 test accuracy.

Figure 4: **Comparison with MLPMixer [14] and ViT. [12]** We compare our method with MLPMixer and ViT. on the CIFAR10 (4a) and on the CIFAR100 (4b) datasets. Our method significantly outperforms both architectures in the low-data regime.

0.9, weight decay $5e{-}4$, and divide the learning rate by $10$ after $80\%$ epochs. We used cross-entropy loss as supervision. For the more complex Caltech datasets, we start with an Imagenet-pre-trained backbone and reduce the dimensionality in our FR only to $256$. We use the same training setup for a full fine-tuning, except that we reduce the initial learning rate to $1e{-}3$ and train for only $100$ epochs.

### 3.1 Supervised Learning

**Comparisons with ResNet18 [8].** We compare the results of our method with those of ResNet18. As shown in Figure 3a, on the first training cycle (1000 labels), our method outperforms ResNet18 by 7.6 percentage points (*pp*). On the second cycle, we outperform ResNet18 by more than $10pp$. We keep outperforming ResNet18 until the seventh cycle, where our improvement is half a percentage point. For the remaining iterations, both methods reach the same accuracy.

On the CIFAR100 dataset (see Figure 3a), we start outperforming ResNet18 by $5.7pp$, and in the second cycle, we are better by almost $7pp$. We continue outperforming ResNet18 in all ten cycles, in the last one being better by half a percentage point. We see similar behavior in Caltech101 (see Figure 3c) and Caltech256 (see Figure 3d).

A common tendency for all datasets is that with an increasing number of labeled samples, the gap between our method and the baseline shrinks. Therefore, dropping the fully-connected layers in case of a large labeled dataset does not cause any disadvantage, as was found in [6]. However, that work did not analyze this question in the low-data regime, where using FC layers after CNN architectures is clearly beneficial.

**Comparisons with MLPMixer [14] and ViT [12].** We also compare the results of our method with that of two non-convolutional networks, the MLPMixer and ViT. Similar to us, the power of MLPMixer is on the strengths of the fully-connected layers. On the other hand, ViT is a transformer-based architecture. Each Transformer block contains an attention block and a block consisting of fully-connected layers. Unlike these methods, our method uses both convolutional and fully-connected layers to leverage the advantages of both the high-level convolutional features and the global interrelation from the fully-connected layers. In Figure 4a, we compare to MLPMixer and ViT on the CIFAR10 dataset. On the first cycle, we outperform MLPMixer by $4.4pp$ and ViT by circa $6pp$. We keep outperforming both methods in all other cycles, including the last one where we do better than them by circa $13pp$. As we can see, MLPMixer and ViT do not perform well even in the latest training cycles and *require to be trained on massive datasets.* We show a similar comparison for CIFAR100 in Figure 4b.

**Comparisons with Knowledge Distillation baselines.** To show that our method's main strength comes from our FR head, we compare to several knowledge distillation (KD) methods. DML [21]

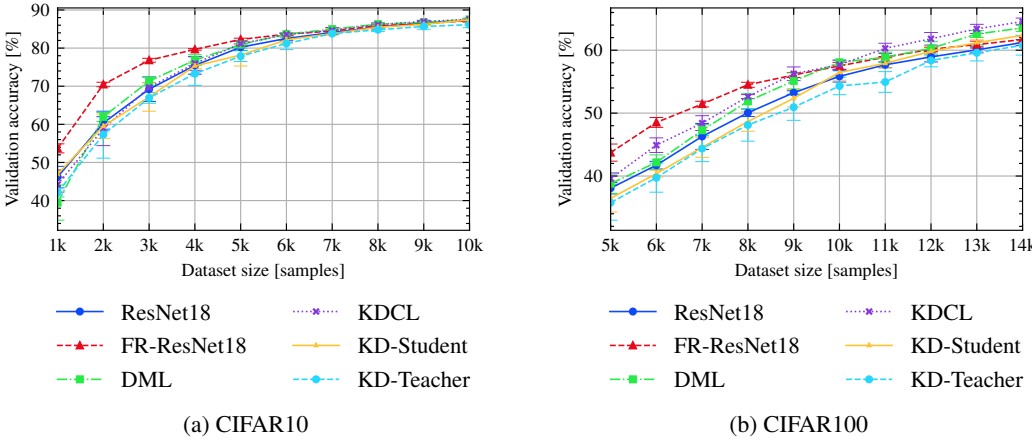

(a) CIFAR10               (b) CIFAR100

Figure 5: **Comparisons with Knowledge Distillation baselines.** We compare to several KD methods. Our method outperforms the baselines in the earlier iterations with a large margin.

trains two networks in an online KD setting. KDCL [22] aims to improve the generalization ability of DML by treating every network as students and training them to match the pooled logit distribution. Finally, KD [23] is the original offline method, which trains a teacher network, then distills its knowledge into a student. In each experiment, the teacher network is ResNet50, the student is ResNet18. As can be seen in Figure 5, our method significantly outperforms the KD methods, by up to 8.5pp on CIFAR10 and 3.6pp on CIFAR100 in the second iteration. In the later iterations, when more data is available and our approach converged to the baseline network's performance, we start to see the benefit of the other KD methods. However, while our method comes with only $0.37pp$ extra parameters during training, the other KD methods use $210.5pp$ more parameters.

**Comparison with SimSiam [24].** Similarly to our method, SimSiam [24] uses a stop-gradient technique. They train a feature extractor in a self-supervised setting, then fine-tune a classifier head in a supervised setting. While SimSiam [24] can also be directly trained on smaller datasets, its behavior on the low data-regime has not been investigated yet. We compare our method against the CIFAR10 version of SimSiam on our datasplits and evaluate its performance with a kNN and with a linear classifier. The training scheme follows SimSiam [24], Appendix D. As it can be seen in Figure 6, our method significantly outperforms SimSiam [24] in all data splits by a maximum margin of $25.45pp$ in the second iteration, and by $8pp$ in the last iteration.

## 3.2   Active Learning

We put our method to the test under a typical low-data setting, active learning, where instead of randomly choosing samples to label, we choose them based on an acquisition score. The results in Section 3.1 can be considered a special case of active learning with a random acquisition score. We use maximum entropy acquisition score with our network and compare it with plain ResNet18 with maximum entropy and two state-of-the-art active learning methods, LLAL [20] and core-set [25].

In Figure 7, we summarize the results on all four datasets, showing our method's superior performance. On CIFAR10, we outperform all methods in early cycles by a large margin, up to $7.3pp$ compared to the core-set approach in the second cycle. As more labels become available during training, all the methods tend to converge to the same result. A similar trend can be observed on the CIFAR100 dataset. Here, our approach achieves $5.2pp$ better accuracy in the second cycle. On the Caltech101 dataset, we have a slight advantage ($< 0.5pp$) compared with the vanilla ResNet18. We have a larger advantage on the Caltech256 dataset, where we tend to outperform the second-best method by $1-1.5pp$. Furthermore, note that core-set and LLAL approaches perform much worse on these more complex datasets. The final gap between the maximum entropy acquisition and LLAL is $4.5pp$ on the Caltech256 and $1.4pp$ on the Caltech101 dataset.

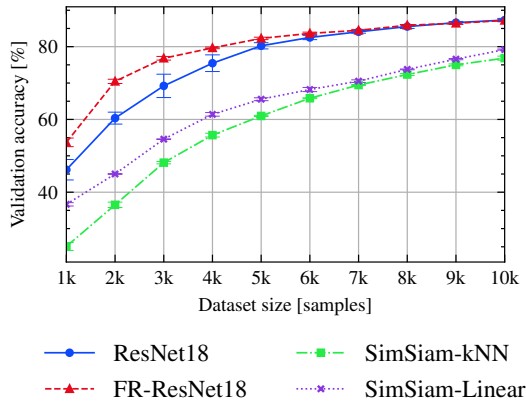

Figure 6: **Comparison with SimSiam [24].** We compare our method against the CIFAR10 version of SimSiam [24]. Our method significantly outperforms SimSiam in the low-data regime.

## 3.3 Semi-Supervised Learning

We now do an experiment where we present preliminary results in semi-supervised learning. We choose to couple MixMatch [26] with our method and compare it with the original version of MixMatch. We use ResNet18 for the experiment and train the methods using only 250 labels. MixMatch reaches 91.07 accuracy, while our method reaches 91.56 accuracy, improving by half a percentage point, and showing that our method can be used to improve semi-supervised learning.

## 3.4 Backbone Agnosticism

We check if our method can be used with other backbones than ResNet18. We show the results for ResNet34, DenseNet121, and EfficientNetB3 on CIFAR10 and CIFAR100 in Figure 8. The goal of the experiment is to show that our method is backbone agnostic and generalizes both to different versions of ResNet as well as to other types of convolutional neural networks. As we can see, our method significantly outperforms the baselines on both datasets and for all three types of backbones.

## 3.5 Ablation

**Feature Refiner** We conduct a detailed ablation of the architecture of our Feature Refiner, showing the effect of the specific elements. We start with the backbone ResNet18 and build our Feature Refiner up step-by-step. Figure 9a shows the results on the CIFAR10 dataset. First, we apply only a single linear layer without any activation function before the output layer (512x512 w/o Activation). Interestingly, we can already see a great improvement of $4.9pp$ in the first cycle, reaching $5.5pp$ in the second cycle. Second, we change the previously applied linear layer to reduce the dimension, as we did in our Feature Refiner (512x64 w/o Activation). This step further improves the results up to $2.3pp$ in the third cycle. Third, we add our fully-connected layers with the ReLU activation (FR w/o LayerNorm), which yields an improvement of $3.1pp$ in the second cycle. Finally, we use our complete architecture by applying the normalization layers, reaching the best results in the earlier stages, giving an additional performance boost of $1.2pp$ in the second stage.

**Online joint knowledge distillation** We now evaluate our proposed OJKD. In these experiments, we highlight that during inference, we can drop the Feature Refiner head without any noticeable performance loss. We use the same training strategy as described in Section 2.1. We train together with the original and Feature Refiner heads. However, during inference, we evaluate both heads. In Figure 9b we show both heads' performance on the CIFAR10 dataset. As we can see, both heads have barely distinguishable performance, showing that the knowledge of our teacher FR head can be properly distilled into the student original head. Furthermore, we ablate the effect of our proposed Gradient Gate. Without our gating mechanism, we can still improve upon the baseline up to $4.5pp$, but using it can give us a further $5.6pp$ improvement in the second stage.

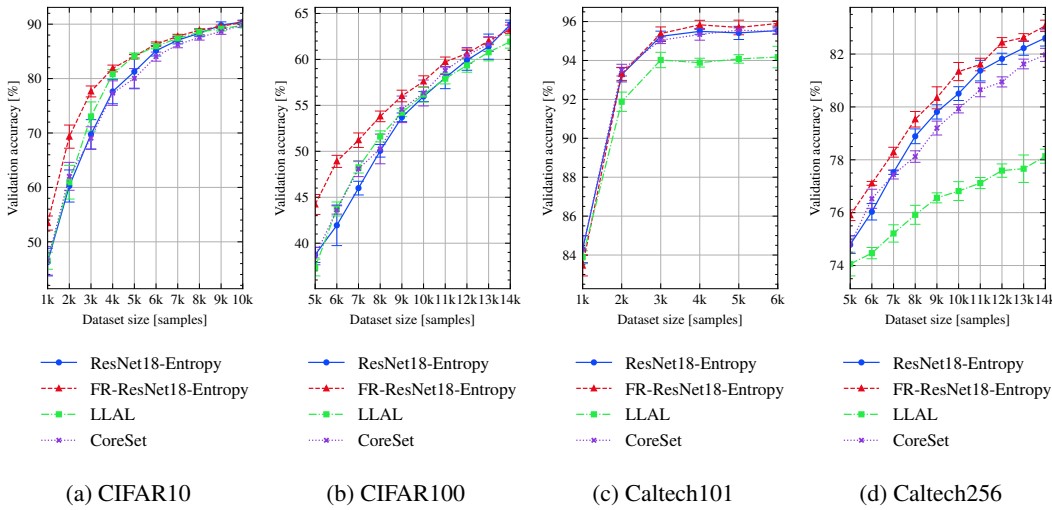

| (a) CIFAR10 | (b) CIFAR100 | (c) Caltech101 | (d) Caltech256 |

Figure 7: **Active Learning.** We compare our approach (FR) in an active learning setup based on maximum entropy. We compare with ResNet18 using entropy-based acquisition score, LLAL, [20] and core-set [25]. Our method significantly outperforms all others, especially in the more challenging earlier stages, when we have a small amount of data.

**The effect of the number of layers.** We now study the number of layers with nonlinearities. To do so, we add nonlinear layers in our Feature Refinement. We show the results on the CIFAR10 dataset in Figure 9c. As we can see, adding more nonlinear layers comes with a decrease in performance. In fact, the more layers we add, the larger is the decrease in performance compared to our original model. This makes sense considering that by increasing the number of layers, we increase the number of learnable parameters, and thus we might cause overfitting.

## 4 Related Work

**All-convolutional networks.** Convolutional neural networks [1, 2] have made a breakthrough in many computer vision tasks since the pioneering work of [4]. Other works [27, 5] improved over those architectures, typically by increasing the number of convolutional layers but by keeping the same architecture: many convolutional and pooling layers followed by a few fully-connected layers. However, this design choice was questioned in the famous *Striving for simplicity* [6] paper, where it was argued that there is no need for fully connected layers. Fully convolutionized architectures such as Google-Inception [7] or ResNets [8] followed, significantly outperforming the previous state-of-the-art. Some more recent works typically tended to improve the architecture of ResNet architectures either by enlarging the number of convolutional filters [8] or densely connecting convolutional layers [10]. Probably the best solution was found in the EfficientNet [28], which simultaneously increases the image resolution, number of convolutional channels, and number of convolutional layers. A constant of these approaches is that they completely get rid of the fully connected layers. Contrary to them, we find out that adding fully-connected layers comes with a massive benefit in performance when the number of labeled points is limited. By doing so, we are able to improve the performance of several backbones in different learning tasks.

**Non-convolutional networks.** A parallel line of research has shown that convolutional neural networks can be replaced either by vision transformers [12, 13] or multi-layer perceptrons [14–16]. The main idea behind these works is to leave the network as much freedom as possible during the learning procedure instead of injecting inductive bias into the network. Indeed, in multi-layer perceptrons, every unit is connected to all the units in both the previous and the next layer, allowing it to use all the information of the proceeding layer as well as pass it into the next one. Works like MLPMixer [14] and others have shown promising results in several vision tasks. Still, they come with a series of limitations, such as the need for pre-training on very large datasets and requiring a massive amount of computational resources. Our work is similar to MLPMixer [14] in the sense that we also leverage the power of multi-layer perceptrons to solve different vision tasks. However, unlike them,

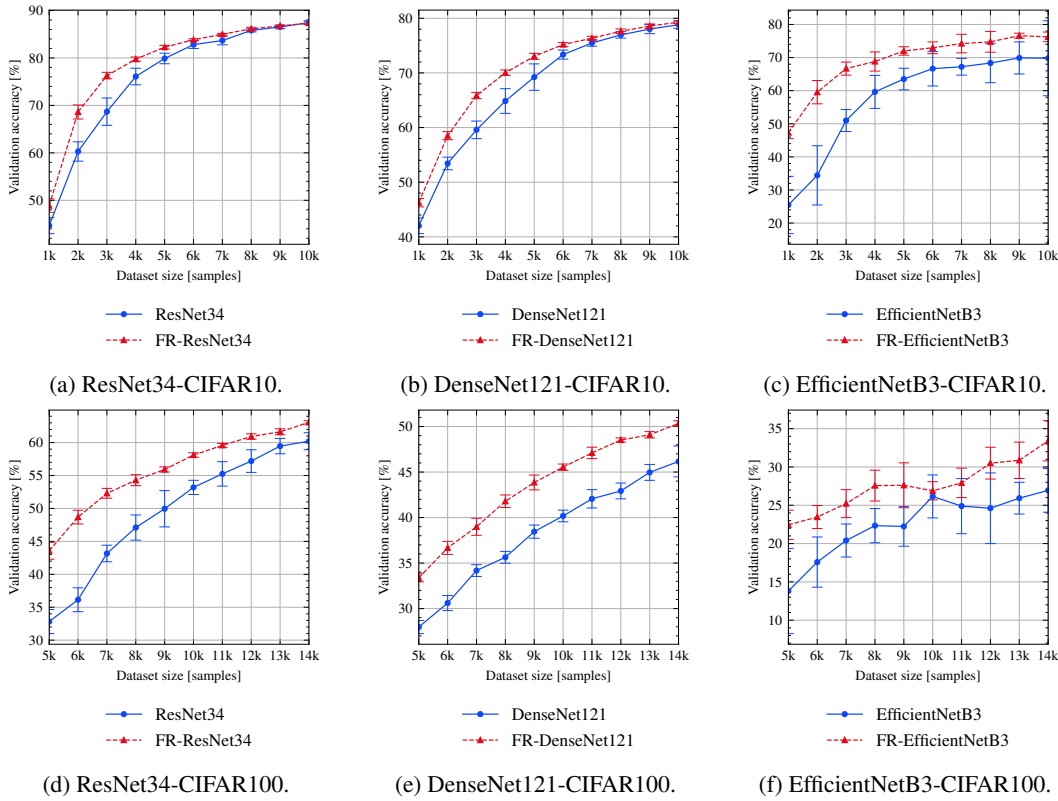

| | | |
|---|---|---|
| (a) ResNet34-CIFAR10. | (b) DenseNet121-CIFAR10. | (c) EfficientNetB3-CIFAR10. |
| (d) ResNet34-CIFAR100. | (e) DenseNet121-CIFAR100. | (f) EfficientNetB3-CIFAR100. |

Figure 8: **Backbone agnosticism.** We show that our approach is agnostic to the backbone. We evaluate our approach on the CIFAR10 and CIFAR100 datasets with ResNet34, DenseNet121, and EfficientNet-B3 backbones. In all cases, our method significantly improves over the original network.

we do not get rid of the convolutional layers, but we advocate for hybrid networks. Our networks do not need pre-training and massive computational resources, while it manages to significantly improve state-of-the-art results when trained with a limited number of labeled points.

**Knowledge distillation.** With the advancements in computational power, larger neural networks are possible to train. However, effective neural networks are required during inference in many practical applications, especially mobile or embedded applications. This requirement motivates the field of Knowledge Distillation (KD). The goal of KD is to reduce the computational demand of a trained network's inference by maintaining its performance. This could be in the form of offline distillation, where one network is used as a teacher to distill its knowledge into another smaller network, the student [23]. However, online distillation has gained more attention lately, thanks to its simpler pipeline [29]. In such a setup, the student and teacher models are trained together end-to-end. For more details on this field, we refer to the work of Gou et al. [29] The published online distillation methodologies are mainly specifically applied for ensemble distillation [30]. On the contrary, we propose an effective yet general online distillation method. We attach a separate head during the training (the Feature Refiner) while using a gating mechanism that blocks the gradients coming from the original softmax layer of the network. On inference, we can simply drop the Feature Refiner head without noticeable performance drop, thus, using no extra parameters.

**Stop-gradient.** SimSiam [31] and BYOL [32] train two branches jointly in a self-supervised setting with different augmentations. In BYOL, the target is provided by the averaged version of the main network, while in SimSiam the same network is used with the stop-gradient technique. These methods share similarities with our OJKD; however, they cannot be directly applied to supervised learning, and they are trained on large datasets. Instead, we focus on generalization from a low amount of data in a supervised setting.

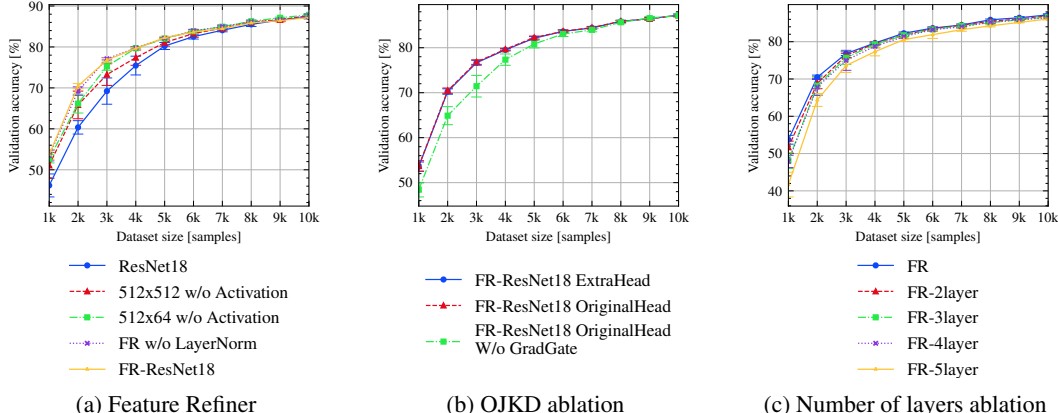

| (a) Feature Refiner | (b) OJKD ablation | (c) Number of layers ablation |

Figure 9: **Feature Refiner** 9a: We apply parts of our Feature Refiner step-by-step. First, we use only a single linear layer without any extra activation (512x512 w/o Activation), then apply our dimension reduction step (512x64 w/o Activation). Finally, we evaluate the effect of the LayerNorm layer.
**OJKD ablation** 9b: Our online joint knowledge distillation enables us to utilize the advantages of our Feature Refiner without increasing the number of model parameters at the same time. Our method also helps even without the Gradient Gate.
**Number of layers ablation** 9c: We study the effect of the number of nonlinear fully-connected layers. More layers do not lead to better performance.

## 5    Conclusion

In this paper, we question the long-believed idea that convolutional neural networks do not need fully-connected layers. Perhaps surprisingly, we show that using hybrid networks that use both convolutional high features and interrelations coming from fully-connected layers improves the generalization performance in the low-data regime, without any drawback in the high-data regime. To show a fair comparison, we also introduce a gating mechanism that allows distilling the knowledge from our extra head with the added FC layers to the baseline network. We show that our approach is model agnostic by evaluating different backbone networks. We show in our experiments that our method yields a significant improvement over the state-of-the-art in supervised and active learning, in a wide range of standard classification datasets without an increase in the number of used parameters.

## Broader Impact / Limitations

A shortcoming of our method is that we cannot improve the networks that already contain fully-connected layers, such as the VGG Network [5]. In the supplementary material, we do an experiment showing that our Feature Refiner cannot further improve VGG Network's accuracy. However, we were able to decrease the number of learnable parameters by $67\%$.

Another deficiency of our paper is the lack of a theoretical explanation. Although the empirical results underline our methodology, we could not find any mathematical proof for it. Our speculative explanation is that if the network has only a single final layer, then the backbone's feature space must be linearly separable. Our intuition is that in the case of a low amount of data, the convolutional layers do not get enough supervision to find the right local reasoning, which limits its flexibility. Having additional final fully-connected layers gives global supervision and increases the flexibility of the network to reach better optima. We note that the research in deep learning has been mostly head by empirical results. We believe that theoretical explanations are desirable and ultimately needed. At the same time papers without a clear (or a downright wrong) theoretical explanation have had a massive impact on deep learning, as is the case of the batch-normalization [33] paper. We hope that our work can open a new interesting line of research, and inspire other researchers to question the 'common knowledge' in deep learning. Furthermore, it would be interesting to see if our results can be generalized in domains where the number of labeled data is very scarce, such as medical imaging.

**Acknowledgements** This work was supported by a Sofja Kovalevskaja Award, a postdoc fellowship from the Humboldt Foundation, the ERC Starting Grant Scan2CAD (804724), and the German Research Foundation (DFG) Research Unit "Learning and Simulation in Visual Computing".

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
