# OpenReview forum: "The Unreasonable Effectiveness of Fully-Connected Layers for Low-Data Regimes"
_NeurIPS.cc/2022/Conference — NeurIPS 2022 Accept_

### Official Review · Reviewer_jkGN · 2022-07-10

**Rating:** 7
**Confidence:** 4
**Soundness:** 2 fair
**Presentation:** 3 good
**Contribution:** 2 fair

**Summary:**

This paper shows that one can improve accuracy in the low-data regime by adding fully connected layers to CNNs during training and distilling knowledge to a classification head, resulting in a network with the same number of parameters and better generalization performance.

**Questions:**

In figure 6f, why does the accuracy of FR-EfficientNet decrease at 10k samples while the accuracy of EfficientNet increases? It seems strange as this effect seems to only happen at this one datapoint.

In figure 4b, 5(c,d), and 6(c,d,e,f) the FR method has higher accuracy than the baseline for all amounts of data; did you run experiments where these networks are trained on the entire dataset? Is there a general point at which there are diminishing returns in accuracy?

**Update**

After the authors provided new experiments to show that their method does in fact outperform knowledge distillation methods in low data regimes, I am satisfied with the evaluation and have changed my review from reject to accept.

**Strengths And Weaknesses:**

I found the method proposed in this paper very interesting and the results showed clear improvements over the base network. Considering how easy this is to implement, I can see this paper having a large amount of impact to the community, and the experiments showing improvements in the active learning and semi-supervised learning setting show the diversity of this approach. However, the lack of knowledge distillation baselines and theoretical backing has me questioning whether this is any different than a typical teach student distillation approach.

For instance, what would happen if one jointly trained ResNet18 and ResNet50, applied a loss similar to [1]? In this case, the test time network would contain no more parameters, so it seems like a reasonable comparison, and it follows a similar intuition of this paper (train with more parameters and drop them during test-time). Although I am not too familiar with current KD methods, I would be surprised if none of them showed a similar gain in performance, so it would be a necessary baseline to compare this papers method to.

A lesser concern I have is with overfitting. Since this method seems to be training a network with more parameters on a small amount of data, I wonder if performance on robustness benchmarks would be worse than the baseline network.

Overall I think that this paper has the potential to have high impact as it is well written and has a simple, easy to follow method that is effective under several tasks. I am giving it a reject as I don't believe the current experiments are sufficient to prove that this method is more advantageous than other knowledge distillation methods, but I am eager to have the authors quell my doubts with a more thorough evaluation.

[1] Zhang et al. "Deep Mutual Learning"

---

> ### Author Response · Authors · 2022-08-02
> **Comments to reviewer jkGN**
>
> **We thank the reviewer for finding our method very interesting, the results showing clear improvements over the base network, the paper being easy to implement, and consequently seeing the paper having a large amount of impact for the community. Considering the reviewer’s praise, we were somewhat disappointed with the low grade of the reviewer. At the same time, we are thankful to the reviewer for mentioning a couple of missing experiments that will ultimately make the paper stronger. We address the reviewer’s concerns and conduct experiments to compare knowledge distillation methods and evaluate the robustness of our method. Our method compares favorably to the baselines, showcasing the benefit of our approach in the low data regime.**
>
> 1. ***Experiments and comparisons with different Knowledge Distillation methods.***
>
> To show that the main strength comes from the fully-connected layers, we follow the reviewer’s recommendation and compare with different knowledge distillation modules. We present a comparison with the suggested Deep Mutual Learning [21] module. Furthermore, we also compare with its follow-up (KDCL [22]) and with the original knowledge distillation method (KD [23]). For all experiments, we use ResNet50 teacher/mutual network for our ResNet18 student network. We have added this discussion and experiments to the manuscript (Figure 5, lines 150-160). For convenience, please click [Figure 5](https://ibb.co/2Pns5Rb), which redirects to the figure which is identical to the one in the revised version of the manuscript.
>
> As can be seen, our method (FR-ResNet18) significantly outperforms the knowledge distillation methods, by up to 8.5pp on CIFAR10 and 3.6pp on CIFAR100 in the second cycle. Furthermore, while our method comes with only 0.37% extra parameters during training (and no extra parameters during inference), the other knowledge distillation methods use 210,5% more parameters. Thus, our method both performs better, and is significantly faster (using one network instead of two).
>
> 2. ***Experiments on robustness benchmarks.***
>
> We extended our paper with a new set of evaluations on robustness benchmarks. We used the CIFAR10-C dataset, which contains a total of 95 perturbed test sets from 19 corruption types with 5 severity levels. We did a thorough evaluation of our method and the baseline ResNet18 network on all the test sets in every iteration with 5 different runs. We have added the experiment to the supplementary material (Section C, Figures 2-4). For convenience, please click [Figure 2 supp](https://ibb.co/KwmfFg2), which redirects to the figure which is identical to the one in the revised version of the manuscript.
>
> Our method (FR-ResNet18) consistently outperforms the baseline in cases of lower corruption severity. In cases of higher corruption severity (severity 3 and severity 4), in the very low-data regime, our method significantly outperforms the baseline. However, with more added data, the baseline starts outperforming our method. In the supplementary material, we also show a similar experiment where we agglomerate results by the type of corruption, reaching the same conclusions.
>
> 3. ***In figure 6f, why does the accuracy of FR-EfficientNet decrease at 10k samples while the accuracy of EfficientNet increases?***
>
> This can be explained by the fact that we used fixed data splits over all the experiments. That specific datapoint can be considered as a bad sample for that architecture. A counterexample can be seen on the same figure at 12k, where FR increases rapidly, but EfficientNet decreases slightly.
>
> 4. ***Is there a general point at which there are diminishing returns in accuracy?***
>
> Yes, with more labeled samples the gap between FR method and the baseline network closes.

---

> > ### Comment · Reviewer_jkGN · 2022-08-05
> > **Thank you!**
> >
> > Thank you for the extra experiments! After looking at the new experiments, I am convinced that your method does in fact perform better than knowledge distillation methods in low data regimes. As my enthusiastic review indicated, I quite enjoy the ideas from this paper but was concerned that the gains were simply due to training with more parameters. I will update my score to an accept, great job!

---

> > > ### Author Response · Authors · 2022-08-05
> > > **Thank you!**
> > >
> > > We thank the reviewer for their encouraging response and praise after the rebuttal. Indeed, the original review was enthusiastic, despite the low score. We acknowledge again that the two sets of experiments, especially the knowledge distillation one, have improved the paper and made its results more convincing.
> > >
> > > We are thrilled to see that the reviewer plans to change their score from **Reject (3)** to **Accept**. **We hope that the reviewer can explicitly update the score so it is not missed by the Area Chair and the other reviewers!**

---

### Official Review · Reviewer_DX6o · 2022-07-11

**Rating:** 6
**Confidence:** 4
**Soundness:** 2 fair
**Presentation:** 3 good
**Contribution:** 3 good

**Summary:**

This paper proposes a training method that involves a module plugged into the final features of a backbone as an additional classification head for joint training with the existing classification head. Along with the original head, the newly added head also pass through the softmax for the cross entropy loss; the training is performed with the summation of the two cross-entropy loss. The proposed module is dubbed Feature Refiner (FR), consisting of two fully connected layers followed by the layer normalizations and operating with the gradient gate (GG) right before the classification head. GG acts exactly like a stop-gradient technique, which is widely used in self-supervised learning, to train the original head only based on the output of the frozen backbone, which is the input for the head, and the backbone is trained only with the extra head with FR. At the inference phase, the original backbone with the original classification head is used for the forward propagation. The authors claim that this training method works well on low-data regimes (for me, it is limited to the low-data regimes), which may not be expected. Some experimental results on small datasets, including the CIFAR datasets and the Caltech datasets for supervised learning, active learning (only on CIFAR), and semi-supervised learning (presumably on CIFAR), are provided to show the effectiveness of the proposed method. The authors try to show the universality of the proposed method of training with some backbones, not constrained on ResNets.


**Questions:**

- Why the ImageNet-pretrained model is used only for training on Caltech datasets?


**Limitations:**

- Limitation is provided, but any potential negative social impacts do not seme to be provided.

**Strengths And Weaknesses:**

### Strengths
- This paper is easy to follow.
- The proposed idea looks somewhat interesting.

### Weaknesses
- The main concern with the proposed method is that it seems to have a very similar training pipeline to SimSiam [1], which showed that using the stop-gradient is a key to training a backbone in self-supervised training. Specifically, Feature Refiner (RF) and the classification heads seem to be the predictor and the projector in the method [1, 2], respectively (the order is reversed but would not be a matter from my standpoint). A difference is at the loss, but as the authors claim (in line 83, p.3), assuming the network is trained in a KD way, the proposed method is a supervised SimSiam (with a single-view ). I hardly agree with the authors' claim that the method performs like a KD,  except for using only a single-view image; the training procedure is very close to SimSiam. Therefore, the authors should argue the difference between the proposed method with SimSiam.
- There is no intuition why the proposed method has the benefit of training with small data.
- The experimental setup is somewhat unconvincing; the setup is inconsistent and does not follow the authors' claim of requiring pre-training for the method (in line 234, page 9). The authors specify that they use an ImageNet-pretrained ResNet for the Caltech datasets training, only providing a seemingly inappropriate reason.
- All the training dataset is small, so the experimental verification of the claim is limited. Using small data for training and a small dataset is technically different. Therefore, it would be better to justify the proposed method on a larger scale dataset such as ImageNet with a small data regime for a stronger claim.
- Comparison of the proposed method with MLPmixer and ViT-B16 in Figure 4 seems unfair. MLPMixer and ViT-B16 have the stem of performing non-overlapping patchification for the input, so training them with 32x32 size images in the CIFAR datasets degrades the model accuracy regardless of the size of training data.
- As aforementioned, the authors claim the proposed method is a joint KD. However, the loss is not a straight KD-based loss (e.g., the KLD loss), and training a backbone with the extra head may not leverage the KD concept, in my opinion. Can the authors elaborate on the concept?


### Pre-rebuttal comment
This work presents a training method using the stop gradient technique with an extra FC head for model training in a supervised manner. Except for using a single-view image and supervised loss, the overall training pipeline is quite similar to the previous self-supervised methods [1, 2], so the authors should elaborate on the difference and provide any intuition why the proposed method could work well. Another concern is that the experiments are not convincing because of the inconsistent experimental setups and small-scale experiments. Therefore, I am leaning towards rejection but would like to see the authors' response and the other reviewers' comments for my final decision.

[1] Chen et al, Exploring Simple Siamese Representation Learning, CVPR 2021
[2] Grill et al, Bootstrap your own latent: A new approach to self-supervised Learning, NeuRIPS 2020

---

> ### Author Response · Authors · 2022-08-02
> **Comments to reviewer DX6o**
>
> **We thank the reviewer for finding our proposed idea interesting, and for finding our paper easy to follow. We agree that the reviewer’s suggestions will improve the manuscript. Below, we address the reviewer’s comments.**
>
> 1. ***The method is very similar to SimSiam and BYOL.***
>
> While methods like SimSiam [1] and BYOL [2] have some similarities with our method, they also have significant differences both a) conceptually and b) technically.
>
> a) Those methods are self-supervised methods, tailored at large datasets. In contrast, our method is a fully-supervised method aimed at network generalization in low-data regimes, hence, the two setups are very distinct.
>
> b) SimSiam and BYOL uses more complex losses than our method and work with multi-view images. In this regard, our method is much simpler than theirs (single view, cross-entropy loss). Furthermore, while our knowledge distillation method has some similarity with stopgradient, there are a few key differences: (i) The most important is that our gradient gating is completely optional. We use it only to reduce the number of parameters at inference, but we reach the same results without it, albeit with a 0.37% increase in the number of parameters. (ii) On inference, their MLP serves as a prediction head, while with our knowledge distillation, we do not use the MLP at all.
>
> We have updated the Related Work section in the manuscript with the discussion above (lines 259-264).
>
> 2. ***No intuition why the method works.***
>
> We agree that we did not provide a detailed explanation about our method, just a minimal hypothetical reasoning in our Limitations section. However, we think that our findings could urge further researchers to pay even more attention to this phenomenon and bring the field closer to understanding the generalization of deep neural networks, especially in the underexplored low-data regime.
>
> 3. ***The authors use pretrained ResNet for Caltech datasets.***
>
> For the Caltech datasets, we used a pre-trained model, because it has a much higher dimensionality. While all CIFAR images are of size 32x32, the Caltech samples have various resolutions. Those samples are resized and cropped (Section A) to the ImageNet size (224x224). Furthermore, we wanted to showcase that our method can also work with pre-trained models.
>
> 4. ***All the training dataset is small, so the experimental verification of the claim is limited.***
>
> We consider the Caltech datasets large and complex enough for the verification. The Caltech datasets contain images of much higher (and even various) resolution. Therefore the complexity of these datasets is similar to ImageNet in terms of dimensionality.
>
> 5. ***Comparisons with MLPMixer and ViT are unfair.***
>
> For the MLPMixer training, we used the timm framework (Section A.3), where the original images are upsampled to the ImageNet size, exactly as they fine-tuned the ImageNet-pretrained MLPMixer model in the official paper for the CIFAR dataset.
>
> Also for training the ViT-B16, we first used the timm framework. However, we found that directly training on the CIFAR10 dataset with a smaller patch size can achieve better results [5(supp)]. Therefore we applied that training strategy.
>
> Finally, both papers compared the results also on the CIFAR datasets (Table 2, Avg 5 in MLPMixer [14], and in Table 2 in ViT [12]). Thus, we are using the same datasets as the authors of those papers used, so the comparison is fair.
>
> 6. ***As aforementioned, the authors claim the proposed method is a joint KD…***
>
> Under Knowledge Distillation we consider methods that optimize a smaller network to achieve similar or better performance than a larger network [23,28]. Our method can be seen as a KD method in the sense that the FR head (which is just slightly larger) is the teacher network, whose knowledge is distilled into the original head. In this case, the teacher and student networks share the backbone, but they have different heads.
>
> However, we would like to emphasize that although we consider our OJKD as an additional contribution, this is not the main contribution of our work since our aim is to improve generalization in the low data regime. In order to make our results comparable to the baseline networks, we developed OJKD; however, we see our results without the OJKD as also quite promising since by adding as few as 0.37% extra parameters, we can achieve a significant improvement.
>
> 7. ***Potential negative social impacts***
>
> Our work aims to improve the generalization of deep neural networks, thus we did not consider any negative social impact specific to our work.
>
> *[1] Chen et al, Exploring Simple Siamese Representation Learning, CVPR 2021*
>
> *[2] Grill et al, Bootstrap Your Own Latent - A New Approach to Self-Supervised Learning, NeurIPS 2020*

---

> > ### Comment · Reviewer_DX6o · 2022-08-07
> > **Response to authors**
> >
> > I thank the authors for the concrete responses; the newly added robustness evaluation was very interesting. Although my concerns have not been fully addressed, I am now inclined to accept because the revised paper contains a few more essential points that adequately reflect the reviewers' concerns. However,  after seeing the response and reviewers, I would like to leave the rest of the concerns that should be reflected in the final paper revision here:
> >
> > > ***1. The method is very similar to SimSiam and BYOL.***
> > While methods like SimSiam [1] and BYOL [2] have some similarities with our method, they also have significant differences both a) conceptually and b) technically.
> > a) Those methods are self-supervised methods, tailored at large datasets. In contrast, our method is a fully-supervised method aimed at network generalization in low-data regimes, hence, the two setups are very distinct.
> > b) SimSiam and BYOL uses more complex losses than our method and work with multi-view images. In this regard, our method is much simpler than theirs (single view, cross-entropy loss).
> >
> > - SimSiam and BYOL mainly experimented with ImageNet and transferred pretrained backbones to small datasets, but it is known that training directly on small datasets works well even. Additionally, the SimSiam paper [1] incorporates the CIFAR experiments in Appendix D. From this standpoint, a similar point of view was pointed out between the proposed method and the methods.
> >
> > - Additionally, I would like to make it clear that the loss of them is not as complicated as the cross-entropy (CE) loss but l2-normalized MSE for SimSiam and BYOL [2]; actually, CE also works for training them [1, 2].
> >
> > > ***3. The authors use pretrained ResNet for Caltech datasets.***
> > For the Caltech datasets, we used a pre-trained model, because it has a much higher dimensionality. While all CIFAR images are of size 32x32, the Caltech samples have various resolutions. Those samples are resized and cropped (Section A) to the ImageNet size (224x224). Furthermore, we wanted to showcase that our method can also work with pre-trained models.
> >
> > - I understood what the authors would like to stress, but I really wanted to know is that whether the Caltech datasets could also benefit from the proposed method without using the ImageNet-pretrained model.
> > Although we don't have much time, can the authors provide some even simple results?
> >
> > > ***5. Comparisons with MLPMixer and ViT are unfair.***
> > For the MLPMixer training, we used the timm framework (Section A.3), where the original images are upsampled to the ImageNet size, exactly as they fine-tuned the ImageNet-pretrained MLPMixer model in the official paper for the CIFAR dataset.
> > Also for training the ViT-B16, we first used the timm framework. However, we found that directly training on the CIFAR10 dataset with a smaller patch size can achieve better results [5(supp)]. Therefore we applied that training strategy.
> > Finally, both papers compared the results also on the CIFAR datasets (Table 2, Avg 5 in MLPMixer [14], and in Table 2 in ViT [12]). Thus, we are using the same datasets as the authors of those papers used, so the comparison is fair.
> >
> > - Thank you for clarifying the training setup. However, I still have the same concern about training MLPMixer and ViT with the same image size in ImageNet pretraining. Because I found that ResNet18 was modified (at the stem and the final GAP) to train with 32x32 images for the CIFAR dataset as confirmed in your provided code, but MLPMixer and ViT were not. Why weren't they customized like that? I don't think the author should follow the ImageNet architectures for MLPmixer and ViT because they don't utilize the ImageNet pre-trained models and finetune them to the CIFAR dataset.
> >
> > - In fact, there are many publicly available MLPMixer and ViT architectures for CIFAR with smaller patch-sizes architecture or something else; therefore, I raised my concerns about the fairness of comparing them in the same arena.
> >
> >
> > **Typos**:  The subfigures in Figure 7 seem to have the same legend.
> >
> >
> >
> > [1] Chen et al, Exploring Simple Siamese Representation Learning, CVPR 2021
> >
> > [2] Grill et al, Bootstrap Your Own Latent - A New Approach to Self-Supervised Learning, NeurIPS 2020

---

> > > ### Author Response · Authors · 2022-08-09
> > > **Comments to reviewer DX6o (second round)**
> > >
> > > **We thank the reviewer for the positive feedback, that they found our new experiments interesting, and that they are now inclined to accept the paper. We conducted additional experiments to address the reviewer’s remaining concerns. We will provide further discussions about these points in the final paper revision.**
> > >
> > >   1. **The method is very similar to SimSiam and BYOL.**
> > >
> > > While SimSiam [30] can also be directly trained on smaller datasets, its behavior on the low data-regime has not been investigated. To better underline this statement, we ran the CIFAR10 version of SimSiam (https://github.com/PatrickHua/SimSiam) on our datasplits and evaluated its performance with a kNN and with a linear classifier (averaged results of 3 runs). The training scheme follows the SimSiam [30] paper, Appendix D. Please click [SimSiam comparison](https://ibb.co/swSsbqP) to see the results.
> > >
> > > As it can be seen, our method significantly outperforms SimSiam [30] in all data splits.
> > > We mentioned that our loss is simpler, considering that we use the default classification loss (cross-entropy), and so our training regime is identical to training a ResNet. However, we agree with the reviewer that the losses used in SimSiam [30] / BYOL [31] are relatively simple too.
> > >
> > > Additionally, we would like to highlight some further technical differences. SimSiam [30] being a self-supervised method claims that their method works because of the self-supervision scheme (Section 5.1: “SimSiam is an implementation of an Expectation-Maximization (EM) like algorithm”), and they derive the EM-like objective from their loss function. This is different from our supervised learning loss. Furthermore, there are other differences like them using multi-view images, and the stopgrad there being crucial (while gradient gating mechanism being optional for us).
> > >
> > >   3. **The authors use pretrained ResNet for Caltech datasets.**
> > >
> > > The other experiments took longer than expected, so we were able to run Caltech101 experiments only today. We show the preliminary results (the first three cycles) in the graphs below. Please click [Caltech w/o pretraining](https://ibb.co/6sM7wvG) for the plot. For convenience we also show the results with ImageNet pretraining. Each experiment was run 5 times, we plot the mean and standard deviation.
> > >
> > > In the 1st cycle, we are worse than ResNet18 by 1pp (with ImageNet pretraining, we were worse by 0.9pp). In the 2nd cycle, we outperform ResNet18 by 0.7pp (with ImageNet pretraining we outperformed them by 0.5pp). In the 3rd cycle, we outperform ResNet18 by 1.4pp (with ImageNet pretraining we outperformed them by 0.8pp). In this way, we show that while both ResNet18 and our method reach significantly lower results than when we use ImageNet pretraining, the relative improvement of our method compared to ResNet18 remains.
> > >
> > >   5. **Comparisons with MLPMixer and ViT are unfair.**
> > >
> > > We thank the reviewer for further clarifying this point. Indeed, we used customized ResNet (both for ResNet18 and FR-ResNet18) for the CIFAR experiments, as in the work of LLAL [20] (we are glad the reviewer checked the code!). For the ViT training, we also used customized versions, as described in [5(supp)].
> > > For the MLPMixer, in the main paper, we used the original architecture. However, based on the reviewer’s advice, we now conducted more experiments and compared multiple customized architectures. We found that MLPMixer-Nano (https://github.com/omihub777/MLP-Mixer-CIFAR) performs better than the original version in CIFAR datasets. However, it still stays behind our method with a large margin (over 10pp in the second, and over 15pp in the third cycle). We have updated the manuscript with the new results. For convenience, please click [Fig MLPMixer](https://ibb.co/pzBPrJY), which points to a figure comparing our previous and new MLPMixer results.
> > >
> > > **Typo: Fixed in the latest revision.**

---

> > > > ### Comment · Reviewer_DX6o · 2022-08-09
> > > > **Thank you for the response**
> > > >
> > > > Thank you for the response! The authors did a great job of dealing with all the concerns, and I am satisfied with most of the new experiments. In the SimSiam experiment, there is, of course, a clear gap in performance between training with the GT labels in a supervised manner and self-supervised learning. Therefore, I recommend the authors compare them in a pre-training and fine-tuning regime, but I don't think this comparison is within the scope of this paper. For the other experiments, It seems that new experimental results have not been updated in the paper yet, but I hope the authors add these to make the paper more solid in the final revision. I will update my score and vote for acceptance of the paper.

---

> > > > > ### Author Response · Authors · 2022-08-09
> > > > > **Thank you**
> > > > >
> > > > > We are glad to see that all the reviewer's concerns have been resolved and that the reviewer increased the score of the paper.
> > > > > We assure the reviewer that we will update the paper with the new experiments for the final revision.

---

### Official Review · Reviewer_yAFj · 2022-07-11

**Rating:** 6
**Confidence:** 3
**Soundness:** 3 good
**Presentation:** 4 excellent
**Contribution:** 3 good

**Summary:**

The authors investigate the value of fully connected layers at the end of convolutional neural networks in the small data regime. They demonstrate that the addition of these layers significantly improves model quality in this regime.

**Questions:**

1. I’d like for the authors to include data on the % increase in parameter count with their mechanism in all experiments, as I stated above.
2. I’m confused by the author’s characterization of ResNet and EfficientNets as fully convolutional. These architectures are typically drawn with a single fully connected layer following the convolutional component of the model.
3. Caltech101 in Figure 3 appears to be a counterexample. I’m wondering if you can explain this more. Your explanation claims you see similar behavior on this dataset but that is not what I see looking at Figure 3c
4. One ablation that would be interesting is only training an FC stack on the target task and showing its quality relative to the results in Figure 3. I am curious if it would outperform on very small datasets and then be surpassed as the dataset grows.
5. I was not able to understand the sentence “We show in our experiments that the network with the train time added fully connected layers still significantly outperforms the original architecture, even if both have an equal number of weights”. I’d like for the authors to clarify this in the text.
6. Some numbers called out in the text are very specific when I would expect them to vary depending on the target architecture. For example, the “512” and “64” in Figure 1 and the “42k” in section 2. I think the text might be more clear if the authors described these dimensions in abstract and stated their exact parameterization in the experiment details.


**Limitations:**

The above sections detail limitations/questions I’d like to address.

**Strengths And Weaknesses:**

Strengths
1. I found the paper to be very clear and well written. It was easy to understand what the authors were doing and why.
2. The results seem significant. This is not a phenomenon that I was aware of previously, although I am not an expert in the use of deep learning in the small data regime.

Weaknesses
1. Distillation is known to improve quality for a constant parameter count so I’m not convinced that the distillation experiments disprove the hypothesis that the quality gains of adding fully connected layers are from increased parameter count. A more convincing argument is that you’re not increasing the parameter count much because of the dimensionality reduction in your FC stack. I’d encourage the authors to include data on the % increase in model parameters with their proposed addition.

---

> ### Author Response · Authors · 2022-08-02
> **Comments to reviewer yAFj**
>
> **We thank the reviewer for finding our paper well-written, with significant results and for the overall positive grade. Below, we further clarify the reviewer’s concerns.**
>
> 1. ***Percentage increase in the parameter count.***
>
> We have updated our supplementary material to include the number of extra parameters for all our experiments. We would like to emphasize that these parameters are used only at train time, and not used during inference.
>
> |                   | CIFAR          | Caltech         |
> |-------------------|----------------|-----------------|
> | ResNet18          | 11173962       | 11228325        |
> | FR_ResNet18       | +42058 (0.38%) | +289893 (2.58%) |
> | ResNet34          | 21282122       | x               |
> |    FR_ResNet34    | +42058 (0.20%) |        x        |
> |    DenseNet121    |     6964096    |        x        |
> |   FR_DenseNet121  | +74826 (1.07%) |        x        |
> |   EfficientNetB3  |    10711602    |        x        |
> | FR_EfficientNetB3 | +82660 (0.77%) |        x        |
>
> 2. ***Why are ResNets characterized as fully-convolutional?***
>
> With the term fully convolutional architectures, we mean neural networks which contain no fully-connected layers except the output layer (the classifier). This way, the architectures such as ResNet or EfficientNet are fully convolutional since no non-linear layer is used after the convolutional backbone, except the final linear output layer. This is different to earlier architectures, such as VGG or AlexNet, where  the convolutional backbone was followed by a series of FC layers with non-linearities.
>
> 3. ***Caltech101 in Figure 3 appears to be a counterexample.***
>
> On Figure 3c, our method consistently outperforms the baseline network, except in the first stage. Unfortunately, we do not know the exact reason behind that. One explanation is that some data splits are more beneficial for the baseline network than for our approach, which causes the baseline network to generalize better. However, since this is the only datapoint across many experiments, it shows that our method can utilize the data splits in general better, and we consider that datapoint as a statistical counterexample.
>
> 4. ***Comparing with an FC stack.***
>
> In Figure 4, we compare with MLPMixer, the flagship of fully-connected neural networks. If the reviewer meant something else, we will be happy to discuss this during the reviewers-authors discussion period.
>
> 5. ***Rewriting the sentence “We show in our experiments that the network…”***
>
> We reformulated this sentence for clarification:
> “We show in our experiments that this reduced network achieves the same test accuracy as the larger (teacher) network and thus significantly outperforms the equivalent architecture that does not use our method.”
>
> 6. ***Some numbers called out in the text are very specific when I would expect them to vary depending on the target architecture.***
>
> We introduced variables for those numeric values and defined their values in the experiment section.
>
> 7. ***Experiments on knowledge distillation.***
>
> We conducted new experiments and compared our method (FR-ResNet18) with three knowledge distillation methods. Deep Mutual Learning (DML) [21] and its follow-up KDCL [22] are online methods, KD is the original offline method [23]. In each experiments, ResNet50 was used as teacher/mutual network for the baseline ResNet18.
>
> Our method significantly outperforms the knowledge distillation methods, by up to 8.5pp on CIFAR10 and 3.6pp on CIFAR100 in the second iteration. Furthermore, the other methods use 210,5% more parameters during training. We have added this discussion and experiments to the manuscript (Figure 5, lines 150-160). For convenience, please click [Figure 5](https://ibb.co/2Pns5Rb), which redirects to the figure which is identical to the one in the revised version of the manuscript.

---

> > ### Comment · Reviewer_yAFj · 2022-08-05
> > **Reviewer response**
> >
> > Thank you for your response and updates. The data on the number of additional training parameters shows that the quality improvements you see are likely not a result of this increase. I think this paper is reasonably above the acceptance threshold and it seems the authors did a good job of addressing the feedback from the other reviewers.

---

> > > ### Author Response · Authors · 2022-08-05
> > > **Thank you**
> > >
> > > We thank the reviewer for praising our rebuttal that addresses their and the other reviewers' concerns. We also thank the reviewer for explicitly saying the paper is reasonably above the acceptance threshold.
> > >
> > > We hope the reviewer can champion our paper, and we are willing to further discuss any doubt the reviewer (or other reviewers) have during the review process.

---

### Author Response · Authors · 2022-08-02
**General Comments to All Reviewers/AC**

**We thank all reviewers for their valuable suggestions and constructive feedback.** We are happy that they found the paper “clear and well-written” (yAFj, DX6o, jkGN), the **“results significant”** (yAFj), the proposed method **“very interesting”** (jkGN) and **“easy to implement”** (jKGN), which shows **“clear improvement”** (jkGN), and they can see the potential of **“having a large amount of impact to the community”** (jkGN) .

We would like to highlight that our method is **not only a knowledge distillation method**. Our work aims to improve the generalization of deep neural networks in low-data regimes, and to show how adding fully-connected layers significantly improves the results. Our knowledge distillation is only to show that we can use the original backbone (trained with knowledge distillation), during inference. However, even without knowledge distillation, the method reaches the same results, albeit with a marginal increase in the number of parameters (0.37%).

### Summary of new experiments

Reviewer jkGN found the paper interesting, and having potentially high impact, but scored it most negatively. They had concerns with the lack of comparison with knowledge distillation methods (explicitly mentioning one), and robustness experiments. In the rebuttal, we have **compared our method with 3 knowledge distillation methods** (including the one mentioned by the reviewer), and included **experiments on 95 corrupted CIFAR test sets** to show the robustness analysis. In these experiments, our method compares favorably against the baselines and shows the benefit of our method in the low data regime. Please see Point (1) and (2) under reviewer jkGN for the experiments. For convenience, the knowledge distillation experiment is also provided under Point (7) of reviewer yAFj.

We updated the manuscript based on the reviewers’ suggestions and put the new experiments in the main manuscript and supplementary material.

---

### Meta-Review · Area_Chair_7wFt · 2022-08-27

**Recommendation:** Accept
**Confidence:** Certain

**Metareview:**

The paper shows that using final fully-connected layers helps the generalization of convolutional neural networks in low-data regimes. The addition of these layers significantly improves model quality resulting in a network with the same number of parameters and better generalization performance.

Initially reviewers had mixed evaluation of the paper. All the reviewers saw that the proposed method is simple and easy to follow, at the same time providing clear improvements over baselines. Also agreed that the results are "significant" and "surprising" effect. There were some concerns raised by the reviewers but the author's rebuttal mostly addressed and improved the paper with sufficiently more experiments and analysis supporting the main claim. Reviewer `DX6o` mentioned that there are few updates promised by the authors which can't be validated until camera ready but it does not seem to warrant block publication.

The Author-Reviewer discussion period was active and the authors did a great job clearing various concerns and questions and all reviewers agreed to support acceptance of the paper. The paper demonstrates a simple yet effective method for small data regime which would be interesting to the broad NeurIPS audience both for practitioners as well as researchers.

**Award:**

No

---

### Decision · Program_Chairs · 2022-09-14

Accept